# Aging-Induced Changes in *Cutibacterium acnes* and Their Effects on Skin Elasticity and Wrinkle Formation

**DOI:** 10.3390/microorganisms12112179

**Published:** 2024-10-29

**Authors:** YeonGyun Jung, Ikwhan Kim, Da-Ryung Jung, Ji Hoon Ha, Eun Kyung Lee, Jin Mo Kim, Jin Young Kim, Jun-Hwan Jang, Jun-Tae Bae, Jae-Ho Shin, Yoon Soo Cho

**Affiliations:** 1Burn Institute, Hangang Sacred Heart Hospital, Hallym University College of Medicine, Seoul 07247, Republic of Korea; jyg1076@gmail.com; 2Department of Integrative Biology, Kyungpook National University, Daegu 41566, Republic of Korea; ikwhankim0926@gmail.com (I.K.);; 3R&D Center, Kolmar Korea, Seoul 06800, Republic of Korea; jh_cos@kolmar.co.kr (J.H.H.); kimjinmo@kolmar.co.kr (J.M.K.); jinyoung_cos@kolmar.co.kr (J.Y.K.); 4Department of Rehabilitation Medicine, Hangang Sacred Heart Hospital, Hallym University College of Medicine, Seoul 07247, Republic of Korea; eunlee0617@gmail.com; 5J2KBIO, Chungbuk 28104, Republic of Korea; jjh1117@j2kbio.com (J.-H.J.); jtbae@j2kbio.com (J.-T.B.); 6Department of Applied Biosciences, Kyungpook National University, Daegu 41566, Republic of Korea; 7KNU NGS Core Facility, Kyungpook National University, Daegu 41566, Republic of Korea

**Keywords:** skin aging, skin microbiome, biomechanical properties, metagenomic sequencing, microbial diversity

## Abstract

Skin aging involves biomechanical changes like decreased elasticity, increased wrinkle formation, and altered barrier function. The skin microbiome significantly impacts this process. Here, we investigated the effects of decreased *Cutibacterium acnes* abundance and increase in other skin microorganisms on skin biomechanical properties in 60 healthy Koreans from Seoul, divided into younger (20–29 years) and older (60–75 years) groups. Metagenomic sequencing and skin assessments showed that the older group exhibited decreased *C. acnes* dominance and increased microbial diversity, correlating with reduced skin elasticity and increased wrinkles. In the younger age group, the enriched pathways included zeatin biosynthesis, distinct biotin metabolism pathways, and cofactor and vitamin metabolism in the younger age group, whereas pathways related to lipid metabolism, energy metabolism, and responses to environmental stressors, including UV damage and pollution, were enriched in the older group, according to functional analysis results. Network analysis indicated higher microbial connectivity in the younger group, suggesting a more stable community, whereas the older group’s community displayed higher modularity, indicating more independent and specialized clusters. This study enhances our understanding of the impact of skin microbiome changes on skin aging, particularly the anti-aging effects of *C. acnes*. Future research should focus on the physiological mechanisms of skin microbiota on skin aging and explore therapeutic potentials to enhance skin health.

## 1. Introduction

Skin, the largest organ of the human body, acts as a dynamic interface between internal physiology and the external environment [1]. Central to this interface is the skin microbiome, a diverse community of microorganisms, including fungi, bacteria, archaea, viruses, and microscopic insects like *Demodex* spp., with bacteria being the most predominant [2,3,4]. Recent studies have highlighted that as the skin ages, microbial diversity tends to increase, with key species such as *Cutibacterium acnes* and *Staphylococcus* undergoing significant shifts, which in turn can affect skin barrier function and immune response, potentially accelerating the signs of aging [5]. The composition and function of these microbial communities are crucial for skin health, affecting local and systemic immune responses [1,5]. Skin conditions, such as atopic dermatitis [6], dandruff [7,8], and vitiligo [9], have been linked to the skin microbiome, and they are influenced by host-related factors such as gender and age [10]. The skin microbiome changes in response to aging, a natural and complex biological process affecting various body systems [5,11].

Skin aging is characterized by decreased elasticity, increased wrinkles, and altered barrier function [11,12]. Such changes, while often cosmetic, impact overall skin health and susceptibility to skin diseases significantly. Recently, studies investigating the impact of aging on the skin microbiome have increased, revealing that aging is associated with notable changes in skin microbial diversity and composition, suggesting potential links between these changes and visible signs of skin aging [5,11,12,13,14]. In addition, recent advances suggest that the skin microbiome composition may serve as a biological clock for predicting skin aging, providing new avenues for targeted anti-aging therapies based on microbial communities [15]. Complex interactions between intrinsic factors (genetic factors [12,16], DNA repair, and antioxidant capacity [5]) and extrinsic aging factors, including environmental influences such as UV exposure [5,17,18] and pollution [19], are being explored. For example, UV exposure generates reactive oxygen species (ROS) that regulate gene expression related to collagen degradation and elastin accumulation, leading to photooxidative skin aging and skin cancer [5]. Similarly, pollution can alter the skin microbiome network, adversely affecting skin barrier function and antioxidant capacity [19]. These factors affect the skin’s structural integrity and influence microbial habitats, potentially impacting the aging process [11].

Interest in the role of *Cutibacterium acnes* in skin aging has grown recently [20,21,22]. While primarily recognized for its association with acne, where it can contribute to inflammation and bacterial colonization [23], *C. acnes* also plays a multifaceted role in skin health and aging. The balance of *C. acnes* within the skin microbiome changes with age, potentially influencing skin aging. Research indicates that as *C. acnes* decreases in older adults, the overall skin microbiome diversity increases [13,20,21,24]. *C. acnes* plays a crucial role in skin health by secreting antimicrobial substances, immunomodulatory agents, and short-chain fatty acids. Therefore, the age-related decline in *C. acnes* may reduce these benefits, contributing to clinical and physiological signs of aged skin [13].

Most studies investigating the correlation between skin aging and microbial communities have utilized 16S rRNA amplicon sequencing [13,14,20,25,26,27,28]. However, the method is limited to identifying taxa up to the genus level and lacks the functional gene information necessary for understanding skin microbial community dynamics comprehensively [29,30]. In addition to 16S rRNA sequencing, fluorescence-based methods have proven crucial in studying *C. acnes*, particularly in tracking its colonization and activity on the skin surface [23]. Consequently, recent studies have employed whole-metagenome sequencing for more precise data [21,24,31,32]. Nevertheless, such studies use only subjective assessment of wrinkle grade levels when investigating the correlation between wrinkles and microbial communities and do not include objective assessments such as wrinkle analysis of skin replicas or elasticity [24].

The present study addresses the gaps above by employing whole-metagenome sequencing for species-level identification and functional profiling, alongside objective measurements of skin biomechanical properties like elasticity and wrinkle formation. By integrating both genomic and biomechanical assessments, the present study provides a comprehensive understanding of the role of the skin bacterial community, particularly *C. acnes*, in skin aging. The approach offers in-depth insights into the complex interactions between the skin microbiome and aging, revealing potential pathways for anti-aging strategies and microbiome-targeted therapies.

## 2. Materials and Methods

### 2.1. Subject Recruitment and Sample Collection

This study examined changes in skin microbial communities with age and the correlation between *C. acnes* and skin biomechanical properties. Sixty healthy Korean volunteers in Seoul were recruited to participate in the present study. Participants were recruited through poster advertisements placed in local community centers. Interested individuals were screened via a preliminary questionnaire to ensure they met the inclusion criteria, such as being free from chronic skin conditions and not having used steroids or antibiotics in the past two weeks. Written informed consent was obtained from all participants prior to enrollment. The study design and protocol were reviewed and approved by the Institutional Review Board of Hangang Sacred Heart Hospital (HG2023-020). Participants were divided into two age groups: the younger (ages 20–29, n = 30) and the older (ages 60–75, n = 30). Menstrual cycle status was taken into account for the female participants in the younger group. Skin sampling and biomechanical measurements were conducted during the follicular phase (10 d after the start of menstruation) for consistency. Female participants in the older group were all postmenopausal. Estradiol levels were measured for all participants to ensure hormonal consistency across groups, and participants with estradiol levels outside the normal reference ranges were excluded from the study. The reference ranges used were follicular phase (21–251 pg/mL), postmenopausal (<10–28 pg/mL), and male (11–44 pg/mL). Exclusion criteria included steroid or antibiotic use within the past two weeks, pregnancy or lactation, and chronic conditions such as diabetes or skin diseases that could affect the skin environment. Participants arrived at the laboratory without makeup or skincare products and acclimatized for 30 min under controlled conditions of 22 ± 2 °C and 50 ± 5% relative humidity before measurements began. For sample collection, the nasolabial fold of each participant was swabbed using sterile swabs containing preservatives (0.1% Tween 20, 0.15 M sodium chloride, 0.1% agar), swabbing for at least 3 min. Each swab was placed in a preservative tube and immediately frozen at −80 °C until DNA extraction.

### 2.2. Measurement of Skin Biomechanical Characteristics

The biomechanical characteristics were measured at the same site as the samples. Melanin and erythema levels, sebum, pH, transepidermal water loss related to skin barrier function, extensibility, and elasticity were measured using the E-CUBE 7^®^ Ultrasound Machine (Alpinion Medical Systems Co., Ltd., Anyang, Republic of Korea), Mexameter^®^ MX18, Tewameter^®^ TM 300, and Cutometer SEM 5801^®^ (all from Courage-Khazaka Electronic GmbH, Cologne, Germany). Wrinkle parameters were measured using skin replica analysis. A skin replica of the target area (at least 10 mm × 10 mm) was created and analyzed with the Skin-Visioline VL 650^®^ (Courage-Khazaka Electronic GmbH, Cologne, Germany). All equipment was used according to the manufacturers’ instructions.

### 2.3. DNA Extraction and Metagenomic Sequencing

Genomic DNA was extracted from swab samples using the DNeasy PowerSoil Pro Kit (Qiagen, Hilden, Germany), with modifications, to enhance DNA yield and purity [33]. To achieve the necessary yield for DNA library preparation, multiple displacement amplification was performed using the REPLI-g Single Cell Kit (QIAGEN, Hilden, Germany) [34]. The quantity and quality of the amplified DNA were assessed using the Qubit^®^ 2.0 Fluorometer (Life Technologies, Carlsbad, CA, USA) and the NanoDrop 2000/2000c Spectrophotometer (Thermo Fisher Scientific, Waltham, MA, USA) to ensure suitability for high-throughput sequencing. The MGIEasy FS DNA Library Prep Set and the DNBSEQ-G400RS High-throughput Rapid Sequencing Set (FCS PE100) (MGI, Shenzhen, China) were used for DNA library preparation. The quality of the prepared libraries was evaluated using the Agilent 2100 Bioanalyzer (Agilent Technologies, Santa Clara, CA, USA). The paired-end libraries were sequenced on the DNBSEQ-G400RS platform (MGI, Shenzhen, China) at KNU NGS Core Facility (Daegu, South Korea), targeting 12.5 million reads per sample, each with a read length of 100 base pairs.

### 2.4. Taxonomic and Functional Profiling

The raw sequence reads with low quality or <50 bp were trimmed using SOAPnuke (v2.1.8) [35], and host reads were removed by mapping to GRCh38 using Bowtie2 (v2.5.1) [36]. Taxonomic classification of clean reads was conducted using Kraken2 (v2.1.3) [37] and Bracken (v2.9) [38]. The clean reads were also mapped against the UniRef90 database using HUMAnN 3.0 (v3.8) [39] to identify gene families, which were then re-grouped based on the Kyoto Encyclopedia of Genes and Genomes database for pathway profiling.

### 2.5. Statistical Analysis

All analyses and visualizations were conducted using R software (version 4.3.3). Alpha diversity (Chao1, Shannon, and direct Simpson indices) and Bray–Curtis distance were calculated using the ‘microeco’ package (v1.3.0) [40]. The Wilcoxon rank-sum test, a non-parametric test that does not assume a normal distribution, was used to compare statistical significance between groups. The test was selected because the data did not meet the assumptions of normality, making it a more appropriate method for comparing the two age groups. Beta diversity was assessed based on the Bray–Curtis dissimilarity index, with principal coordinate analysis performed to visualize differences in microbial community composition between groups, using the ‘microeco’ package. The original Bray–Curtis dissimilarity index was introduced by Bray and Curtis [41]. Permutational multivariate analysis of variance with 999 permutations was undertaken to statistically compare the compositions. Differential expression of taxa across groups was identified using linear discriminant analysis effect size with the ‘microeco’ package. Linear regression analysis was performed using the ‘ggpubr’ package (v0.6.0) [42] to evaluate the relationship between *C. acnes* abundance and skin biophysical parameters. Spearman’s correlation analyses were conducted using the ‘corrplot’ package (v0.92) [43] to examine the relationships between differentially expressed taxa and skin biophysical parameters and between differentially expressed taxa and KEGG pathways. Procrustes analysis was performed to compare the taxonomic and functional data using the ‘vegan’ package (v2.6-4) [44]. To analyze the co-occurrence network encompassing bacterial species and functional pathways, a filtering criterion of species and pathway prevalence >80% and an abundance >0.01% in each group was applied. Correlation network analysis was conducted using the igraph (v1.5.1) R package. Significant correlations were determined using Spearman’s correlation coefficient (q < 0.05, |r| > 0.80). The topological properties, including clustering coefficients, density, and modularity, were estimated using the functions transitivity(), edge_density(), and modularity() in the igraph R package [45].

## 3. Results

### 3.1. Study Population

To investigate whether age-related changes in *C. acnes* within the skin microbiome affect skin biomechanical properties, 60 healthy Koreans were recruited and divided into two age groups. The biomechanical characteristics of the nasolabial fold, including melanin, erythema levels, sebum, pH, transepidermal water loss, extensibility, and elasticity, were measured. The results showed significantly higher transepidermal water loss and greater mean depth, length, and area of wrinkles in the older group compared to in the younger group (*p* = 0.032, *p* < 0.001, *p* < 0.001, and *p* < 0.001, respectively). Skin elasticity measures, including final distensibility (*p* = 0.013), gross (*p* < 0.001), net (*p* < 0.001), and biological elasticities (*p* < 0.001), were significantly lower in the older group. However, viscoelasticity was significantly higher in the older individuals (*p* = 0.011) (Table 1). No significant differences were observed in pH, sebum, melanin, and erythema between the two groups.

### 3.2. Age-Related Decline in C. acnes Dominance and Its Impact on Skin Bacterial Diversity

We identified 109 species-level taxa with an average relative abundance of >0.01% across all skin microbiome communities. Among these taxa, three were considered core, accounting for approximately 83% of the average reads. In the older group, *C. acnes* had an average relative abundance of 60.55 ± 35.68%, whereas in the younger group, it was 83.14 ± 22.18%, indicating significantly higher dominance in the younger group (*p* = 0.007) (Figure 1a,b).

The reduced *C. acnes* dominance in the older group led to higher abundance and diversity of typical skin commensals, resulting in greater bacterial diversity (Figure 1c–e). The Chao1 index, estimating species richness, was 78 ± 19 for the older group and 58 ± 20 for the younger group (*p* < 0.001). Similarly, the Shannon index, measuring species diversity, was 1.29 ± 0.99 for the older group and 0.52 ± 0.57 for the younger group (*p* < 0.001). The Simpson index, focusing on evenness, was 0.44 ± 0.29 for the older group and 0.22 ± 0.22 for the younger group (*p* < 0.001). The results suggest a more diverse and evenly distributed bacterial community in the older group.

Principal coordinate analysis using Bray–Curtis distances clearly separated the two groups (*p* = 0.002) (Figure 1f). The intra-group sample distance was greater in the older group (0.74 ± 0.26) due to decreased *C. acnes* dominance and increased presence of other skin commensals, compared to the younger group (0.42 ± 0.27) (Figure 1g).

In-depth analysis using the linear discriminant analysis effect size tool revealed that only *C. acnes* was a biomarker in the younger group, while 22 diverse species, including *Streptococcus salivarius* and *Staphylococcus aureus*, were biomarkers in the older group (Figure 1h). The results indicate that *C. acnes* dominance decreases with age, leading to an increase in the variety of other skin bacterial communities (Figure 1h).

### 3.3. Correlation Analysis Between Group-Specific Biomarker Bacterial Taxa and Skin Biomechanical Characteristics

We observed that the dominance of *C. acnes* decreases with aging, leading to changes in the skin bacterial community composition. The changes were correlated with skin biomechanical properties. *C. acnes* was positively correlated with gross (r = 0.408, *p* = 0.001), net (r = 0.397, *p* = 0.002), and biological elasticities (r = 0.406, *p* = 0.001). However, it was negatively correlated with wrinkle area (r = −0.366, *p* = 0.004), depth (r = −0.264, *p* = 0.041), and length (r = −0.381, *p* = 0.003) (Figure 2). The results suggest that *C. acnes* maintains skin elasticity, inhibiting wrinkle formation. However, other participant characteristics, such as TEWL, pH, sebum levels, melanin, erythema, final distensibility, and viscoelasticity, did not exhibit significant correlations with *C. acnes* (Appendix A).

Figure 3 presents a heatmap showing the correlations between the 22 biomarkers in the older group and skin biomechanical properties. It reveals significant negative correlations with gross elasticity, with *Staphylococcus aureus* showing the highest correlation (r = −0.264, *p* = 0.041). Net elasticity exhibited significant negative correlations with 20 species, excluding *Staphylococcus lugdunensis* and *Streptococcus pneumoniae*. Biological elasticity exhibited significant negative correlations with 21 species, excluding *Streptococcus pneumoniae*.

*Staphylococcus haemolyticus*, *Staphylococcus saprophyticus*, *Staphylococcus lugdunensis*, *Staphylococcus warneri*, *Corynebacterium macginleyi*, *Staphylococcus pasteuri*, *Rhizobium pusense*, *Staphylococcus hominis*, *Streptococcus gordonii*, *Staphylococcus capitis*, *Streptococcus salivarius*, and *Staphylococcus aureus* all exhibited significant positive correlations with wrinkle depth, length, and area.

### 3.4. Correlation Analysis Between Group-Specific Bacterial Taxa and Potential Functions

To understand how the differences in bacterial taxa identified as biomarkers in each group reflect functional potential differences, we analyzed their functional profiles. The taxonomic composition and functional pathways were consistent among participants within each group (Figure 4a), suggesting that samples with similar taxonomic compositions tend to be functionally identical. Additionally, the functional profiles of the two groups showed distinct differences, similar to their taxonomic profiles (Figure 4b).

In the older age group, functions such as energy metabolism, lipid metabolism, photosynthesis, organismal systems, fructose, mannose, phosphonate and phosphinate metabolism, cell motility, RNA polymerase, and aminobenzoate degradation were more abundant than in the younger group. Conversely, the younger group was characterized by functions such as zeatin biosynthesis, biotin metabolism, pantothenate and CoA biosynthesis, lipoic acid metabolism, and the metabolism of cofactors and vitamins (Figure 4c).

Correlation analysis between bacterial taxa identified as biomarkers and potential functions in each group showed contrasting results between *C. acnes*, a biomarker in the younger group, and the biomarkers in the older group (Figure 5). *C. acnes* showed significant positive correlations with 5 of 13 functional profiles. Among these, zeatin biosynthesis (r = 0.790, false discovery rate [FDR]-adjusted *p* < 0.001) had the strongest positive correlation, followed by biotin metabolism (r = 0.754, FDR-adjusted *p* < 0.001) and metabolism of cofactors and vitamins (r = 0.732, FDR-adjusted *p* < 0.001).

Functional profiles that showed positive correlations with *C. acnes* had no positive correlations with the 22 biomarkers of the older group; rather, most showed significant negative correlations. Additionally, *C. acnes* showed significant negative correlations with eight functional profiles, with lipid metabolism showing the strongest negative correlation (r = −0.677, FDR-adjusted *p* < 0.001). All 22 biomarkers of the older group showed positive correlations with this functional profile. These results suggest that decreased *C. acnes* dominance significantly impacts the microbial community’s functional capabilities.

### 3.5. Network Analysis of Ecological Relationships Between C. acnes and Metabolic Characteristics

We conducted a network analysis integrating taxonomic and functional data to explore the ecological relationships between *C. acnes* and metabolic characteristics (Figure 6). Significant correlations were identified, and network properties such as clustering, density, and modularity were analyzed. In the older group, only ‘other glycan degradation’ was positively correlated with *C. acnes*, while several pathways, including ‘mismatch repair’ and ‘*Staphylococcus aureus* infection’, showed negative correlations. In the younger group, positive correlations were observed with ‘pentose and glucuronate interconversions’ and ‘zeatin biosynthesis,’ while negative correlations included ‘*Staphylococcus aureus* infection’ and ‘fatty acid biosynthesis’. Notably, only ‘*Staphylococcus aureus* infection’ was correlated with *C. acnes* in both age groups. The results suggest that changes in microbial community composition can alter microbial roles, indicating functional shifts across different age groups.

Additionally, network characteristics revealed structural differences between the groups. The older group had a clustering coefficient of 0.510, network density of 0.050, and modularity of 0.755, while the younger group had a clustering coefficient of 0.622, network density of 0.086, and modularity of 0.629. The higher clustering coefficient and density in the younger group indicate a more connected and interactive microbial ecosystem. Conversely, higher modularity in the older group suggests a compartmentalized network with distinct functional units, indicating more specialized functions. These structural differences highlight how microbial roles evolve with age, potentially impacting overall microbial community function and skin health across different age groups.

## 4. Discussion

The skin microbiome plays a crucial role in skin health, prompting many studies on its relationship with various skin diseases [46]. As skin microbial communities change with disease and aging, research aimed at understanding these changes has also increased. While previous studies primarily used 16S rRNA sequencing [5,13,14,20,25,26,27,28], whole-genome sequencing techniques have enabled a better understanding of microbial structure and functional potential at the species level [20,24,31,32]. However, understanding the skin microbiome in the context of aging, particularly regarding the correlation between skin aging-related biomechanical properties and the skin microbiome, is limited. Our study focuses on the role of *C. acnes* on changes in the biomechanical properties of the skin with aging and reveals age-related changes in the skin microbiome. These findings contribute to the existing literature on the skin microbiome’s impact on skin health and provide new insights into the complex interactions between the microbiome and skin aging.

The present study found that higher alpha diversity index values indicated greater species richness and diversity in the older group, consistent with previous findings [13,14,20,21,24,25,28]. A decrease in *C. acnes* was observed in the older group, consistent with previous studies [13,20,21,24,31,32]. These diversity changes are likely related to the reduction in *C. acnes* with aging. Linear discriminant analysis effect size revealed that *C. acnes* was the sole biomarker in the younger group, while 22 different microbial species were identified as biomarkers in the older group. *C. acnes* produces the antibiotic cutimycin and short-chain fatty acids, inhibiting biofilm formation [47,48]. Thus, its decrease with aging may allow for an increase in other microbes, resulting in higher alpha diversity.

Our analysis revealed significant correlations between *C. acnes* abundance and various skin biomechanical properties. *C. acnes* was positively correlated with skin elasticity and negatively correlated with wrinkle formation (Figure 2), suggesting a protective role in maintaining skin structure. *C. acnes* produces short-chain fatty acids that boost essential lipids, contributing to skin homeostasis through antioxidant activity [49,50,51,52]. Conversely, the 22 biomarkers identified in the older group showed opposite results to those of *C. acnes* (Figure 3). Recent studies have highlighted that *C. acnes* supports skin health through multiple mechanisms, including the production of short-chain fatty acids like propionic acid [47,48], which enhances skin barrier function and exhibits anti-inflammatory properties [22,49]. *C. acnes* also has immunomodulatory effects that help maintain a balanced immune response, potentially reducing inflammation-related damage associated with aging [13]. Such effects help maintain a balanced immune response, which could reduce inflammation-related damage associated with aging. Furthermore, *C. acnes* has been shown to reduce oxidative stress, a key factor in skin aging, by neutralizing reactive oxygen species, which can otherwise cause damage to skin cells. By reducing oxidative damage, *C. acnes* may protect skin cells from premature aging and maintain overall skin integrity [53]. This combination of protective biochemical effects supports the hypothesis that *C. acnes* plays a crucial role in delaying skin aging and preserving skin elasticity. In addition to age, other factors such as lifestyle and sebum production may play a role in the correlation between *C. acnes* abundance and skin mechanical properties. Previous studies have shown that sebum levels can influence microbial diversity and *C. acnes* abundance significantly, with higher sebum production potentially enhancing the protective role of *C. acnes* in maintaining skin elasticity and preventing wrinkle formation [54]. Moreover, lifestyle factors, including diet, skincare routines, and exposure to environmental stressors, are likely to impact both the skin microbiome and biomechanical properties. Future research should further explore the combined effects of these factors to better understand their contributions to skin aging.

This group included various staphylococci, such as *Staphylococcus aureus*, which can reduce skin elasticity by stimulating protease activity and inducing inflammation [55,56]. The results of the present study suggest that the decrease in *C. acnes* dominance leads to reduced elasticity maintenance, while the proliferation of other bacteria negatively affects elasticity, contributing to skin aging. However, the conclusions are based on correlational data, and thus, causation cannot be definitively established. The observed negative correlation between *C. acnes* levels and the presence of other bacteria may simply reflect the decrease in *C. acnes*, allowing other bacteria to proliferate. Further research is required to clarify the underlying causal relationships. Additionally, skin staphylococci require nutrients like arginine, cysteine, methionine, valine, and aromatic amino acids for survival [57] and break down compounds like triglycerides, diglycerides, monoglycerides, glycerol, and cholesterol to obtain these nutrients from the skin [58]. Such activities likely contribute to the degradation of skin biomechanical properties.

The functional profiling of microbial communities revealed distinct differences between the younger and older groups. The younger group’s microbial community was characterized by zeatin biosynthesis, biotin metabolism, and vitamin metabolism. Zeatin has anti-aging effects on adult skin fibroblasts in vitro [59], and biotin is essential for skin health, with biotin deficiency leading to dermatitis and skin infections [60]. Other vitamins also regulate various biological functions that impact skin health [61]. Conversely, the microbial community in the older group was associated with pathways related to lipid metabolism, energy metabolism, and responses to environmental stressors such as UV damage and pollution.

Network analysis elucidates complex biological systems through interactions within bacterial communities [62]. It provides insights into the ecological relationships between *C. acnes* and other bacterial species. The bacterial network in the younger group exhibited higher connectivity and interactions, suggesting a more stable and resilient community. In contrast, the older group’s network showed higher modularity, indicating more independent and specialized communities. Network analysis highlighted differential ecological relationships between *C. acnes* and various metabolic functions across age groups. Only ‘*Staphylococcus aureus* infection’ was negatively correlated with *C. acnes* in both groups, while other pathways differed. The findings suggest that aging affects correlations among key bacterial species and their functional roles. The structural differences in these microbial networks reflect functional changes and adaptations that could impact skin health with age.

This study offers valuable insights into the relationship between the skin bacterial community and skin biomechanical properties. However, several limitations should be acknowledged. First, the study was conducted on healthy Korean individuals from Seoul, which may limit the generalizability of the results to other populations with different ethnic backgrounds or geographic regions. The sample size of 30 participants per group may also be insufficient to detect subtle differences. Future research should include a larger, more diverse participant pool to enhance the generalizability of the findings. Second, the focus of this study was on the bacterial component of skin bacterial communities, without accounting for eukaryotic members, such as *Malassezia* spp., which play critical roles in skin health and disease [23,63]. Future research should include both bacterial and eukaryotic microorganisms for a more comprehensive understanding of how various microbial communities influence skin biomechanical properties. Third, factors such as diet, cosmetic use, and geographic variations—known to affect skin bacterial communities—were not controlled for in the present study. Such factors could have influenced skin bacterial composition and biomechanical properties. Future studies should control for the variables and apply multivariate analyses to account for their potential impact. Lastly, while the present study focused on structural and functional changes in the skin bacterial community, the precise mechanisms through which microbes like *C. acnes* affect skin biomechanical properties remain unclear. Future research should investigate the mechanisms by which *C. acnes* contributes to skin elasticity and how other microbes impact skin barrier function. Investigations into microbial metabolic products, such as short-chain fatty acids, are essential. Microbiome modulation strategies, including the use of probiotics and prebiotics to enhance beneficial microbes like *C. acnes*, could also be explored as potential interventions for maintaining skin elasticity and reducing wrinkle formation. Studies involving animal models could provide valuable insights into how the skin microbiome influences the aging process.

In conclusion, this study highlights significant age-related changes in the skin microbiome and their impact on skin biomechanical properties. The age-related decrease in *C. acnes* is associated with increased microbial diversity and altered functional profiles, correlating with reduced skin elasticity and increased wrinkle formation. These findings advance our understanding of the complex interactions between the skin bacterial community and aging, offering potential pathways for developing innovative anti-aging products. Future research should explore microbiome modulation to improve skin health and slow down age-related changes.

## Figures and Tables

**Figure 1 microorganisms-12-02179-f001:**
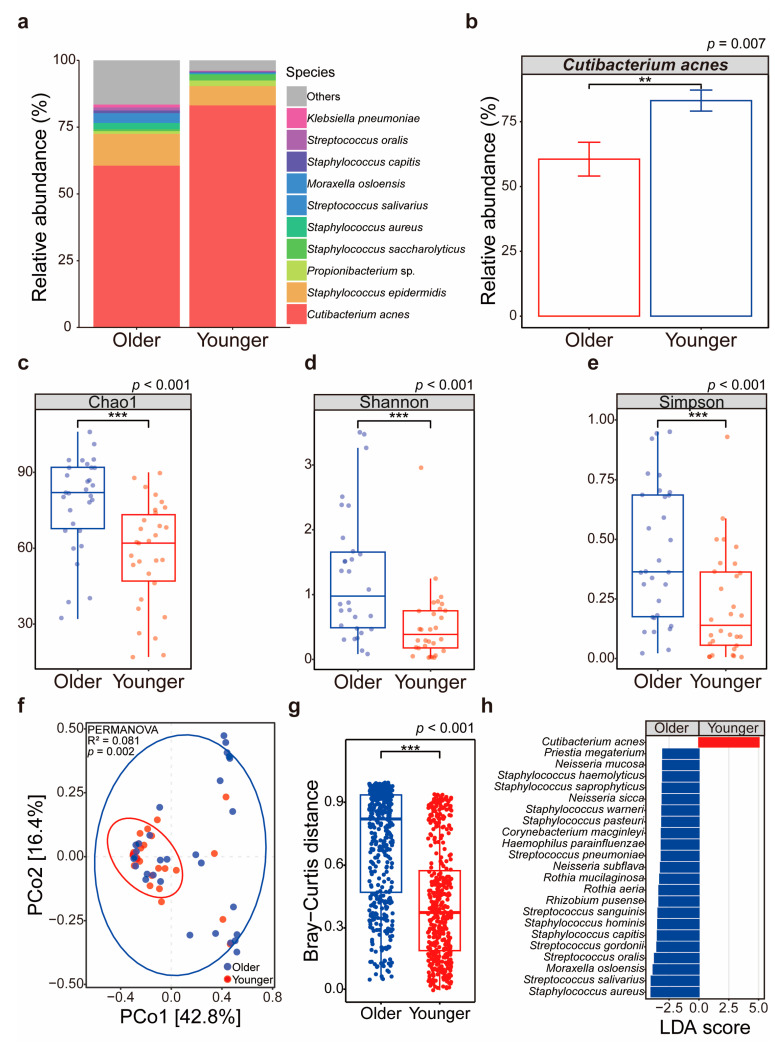
Age-related differences in diversity and dominance of the skin bacterial community. (**a**) Relative abundance of bacterial species in older and younger groups. The color-coded bars represent specific bacterial species. (**b**) Relative abundance of *Cutibacterium acnes* in older and younger groups. (**c**) Chao1 diversity index comparison between older and younger groups. (**d**) Shannon diversity index comparison between older and younger groups. (**e**) Simpson diversity index comparison between older and younger groups. (**f**) Principal coordinate analysis (PCoA) plot based on Bray–Curtis distances, illustrating the separation of bacterial communities between the older and younger groups. (**g**) Bray–Curtis distances within the older group and within the younger group, respectively, illustrating intra-group microbial community differences. (**h**) Linear discriminant analysis effect size identifying significant bacterial biomarkers in older and younger groups. Stars denote the level of significance (unpaired *t*-test; ** *p*-value  <  0.01; *** *p*-value  <  0.001).

**Figure 2 microorganisms-12-02179-f002:**
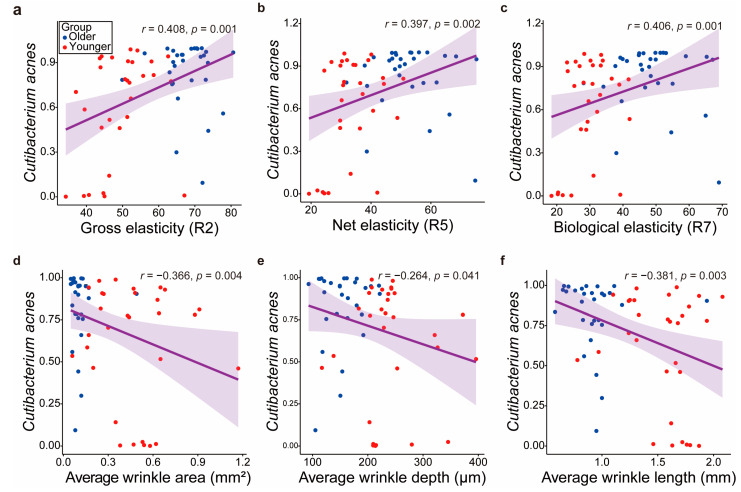
Regression plot for skin biomechanical properties significantly correlated with *Cutibacterium acnes*. Correlation between *C. acnes* abundance and (**a**) gross elasticity, (**b**) net elasticity, (**c**) biological elasticity, (**d**) average wrinkle area, (**e**) average wrinkle depth, and (**f**) average wrinkle length.

**Figure 3 microorganisms-12-02179-f003:**
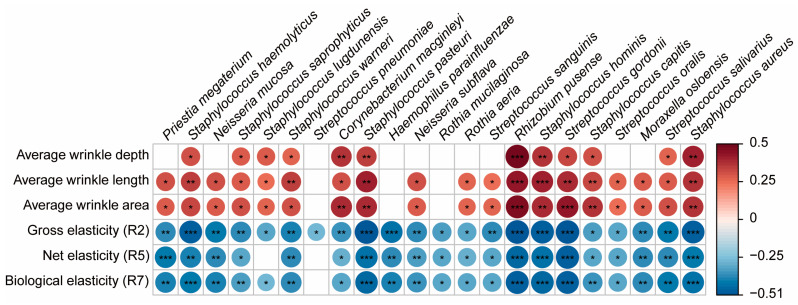
Correlation heatmap of skin biomechanical properties and biomarkers in older individuals. Heatmap of correlations between skin biomechanical properties and biomarkers in the older group. Heatmap illustrating the correlations between the 22 biomarkers identified in the older group and various skin biomechanical properties, including gross, net, and biological elasticities, average wrinkle area, depth, and length. Each cell in the heatmap represents the strength and direction of the correlation, with significant positive correlations in blue and significant negative correlations in red. The biomarkers include species such as *Staphylococcus aureus*, *Staphylococcus haemolyticus*, and *Rhizobium pusense* (* *p* < 0.05, ** *p* < 0.01, *** *p* < 0.001).

**Figure 4 microorganisms-12-02179-f004:**
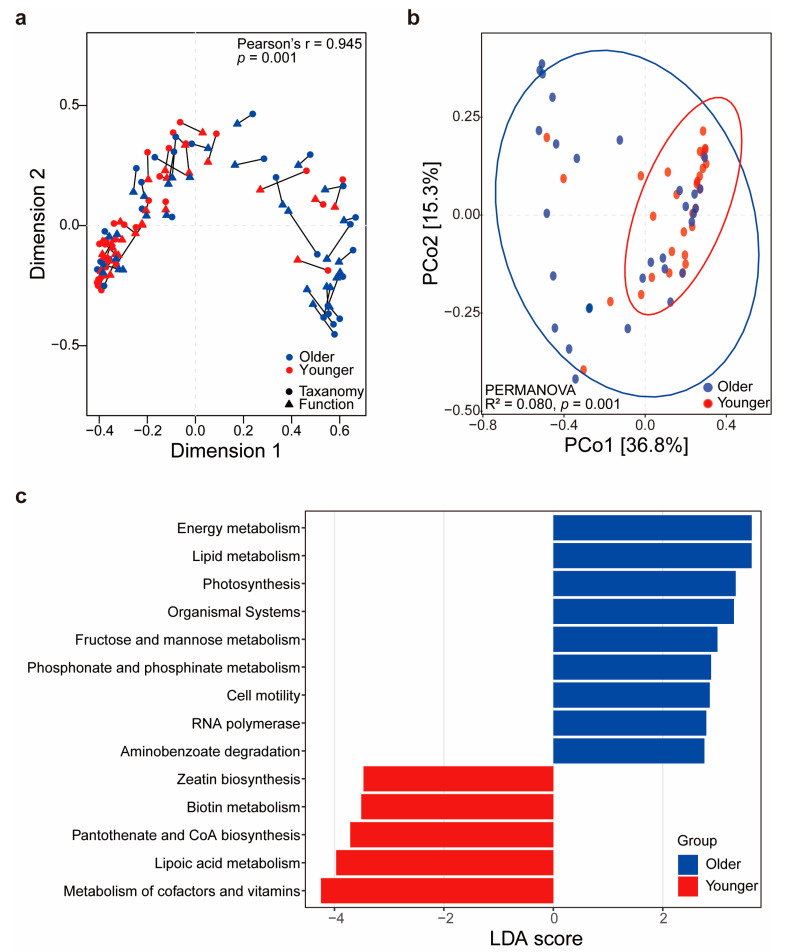
Differences in functional profiles between age groups. (**a**) Procrustes analysis showing congruence between the taxonomic and functional compositions of the microbiome. The statistically significant Pearson correlation indicates that microbiomes had similar taxonomic composition and functions. (**b**) PCoA based on functional profiles, illustrating distinct differences between the younger and older groups. (**c**) Linear discriminant analysis scores of functional pathways enriched in each group.

**Figure 5 microorganisms-12-02179-f005:**
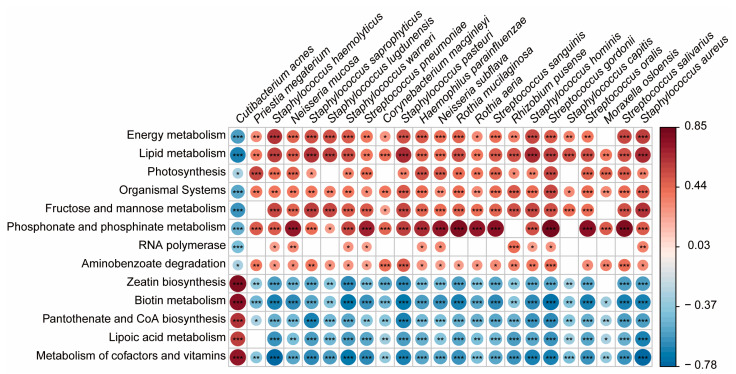
Heatmap of correlations between functional profiles and biomarkers. This heatmap illustrates the correlations between the identified microbial biomarkers and various functional profiles. Positive correlations are shown in red, indicating that an increase in microbial abundance is associated with an increase in the specific functional profile. Negative correlations are shown in blue, indicating that an increase in microbial abundance is associated with a decrease in the specific functional profile. Significance levels are indicated by asterisks (* *p* < 0.05, ** *p* < 0.01, *** *p* < 0.001).

**Figure 6 microorganisms-12-02179-f006:**
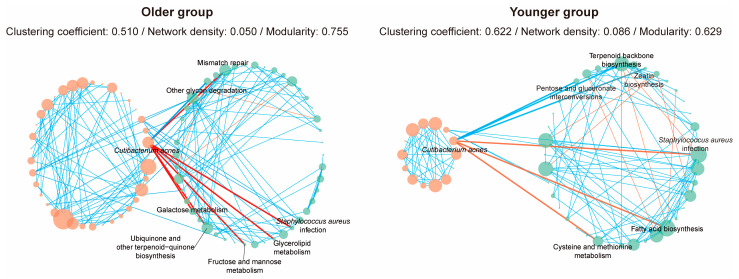
Co-occurrence network analysis of taxonomic and functional profiles in older and younger groups focusing on *C. acnes*. This figure illustrates the co-occurrence networks of bacterial species and functional pathways in older (**left panel**) and younger (**right panel**) groups. Each panel contains two circles: the left circle represents the taxonomic profile, and the right circle represents the functional profile. The size of each node corresponds to its abundance, while the color of the edges indicates the type of correlation—blue for positive correlations and red for negative correlations.

**Table 1 microorganisms-12-02179-t001:** Participant characteristics for age-related study of skin bacterial communities.

Variables		Group		*p*-Value
		Older (*n* = 30)	Younger (*n* = 30)	
Age, years		64.53 ± 4.51	24.83 ± 3.22	<0.001
Sex	Male	15 (50.00%)	15 (50.00%)	>0.999
	Female	15 (50.00%)	15 (50.00%)	
TEWL, g/m^2^/h		17.35 ± 4.25	15.31 ± 2.78	0.032
pH		5.30 ± 0.65	5.39 ±.075	0.630
Sebum, μg sebum/cm^2^		53.80 ± 34.31	75.47 ± 50.87	0.058
Melanin, AU		159.00 ± 47.18	178.20 ± 37.10	0.085
Erythema, AU		261.20 ± 78.73	284.20 ± 66.41	0.227
Average depth of wrinkles, mm		240.50 ± 60.86	153.70 ± 34.34	<0.001
Average length of wrinkles, mm		1.54 ± 0.30	0.92 ± 0.24	<0.001
Average area of wrinkles, mm		0.46 ± 0.25	0.11 ± 0.08	<0.001
Final distensibility (R0), mm		0.11 ± 0.03	0.13 ± 0.03	0.013
Gross elasticity (R2), %		49.30 ± 7.85	67.85 ± 6.13	<0.001
Net elasticity (R5), %		33.44 ± 7.82	52.48 ± 10.57	<0.001
Viscoelasticity (R6), %		12.55 ± 7.23	8.16 ± 5.67	0.011
Biological elasticity (R7), %		29.61 ± 6.20	48.60 ± 9.75	<0.001

Data are expressed as mean  ±  standard deviation or *n* (percentage). The *p*-value was obtained using the Wilcoxon rank-sum test for continuous variables. Statistical significance was set at *p* < 0.05. Abbreviations: TEWL, transepidermal water loss; AU, arbitrary unit.

## Data Availability

Datasets related to this article can be found at https://www.ncbi.nlm.nih.gov/sra/PRJNA1135502 (accessed on 13 July 2024), hosted at Sequence Read Archive under the BioProject accession number PRJNA1135502.

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
