# Peer review of "Aging-Induced Changes in Cutibacterium acnes and Their Effects on Skin Elasticity and Wrinkle Formation"

_microorganisms, 2024, doi:10.3390/microorganisms12112179_

Round 1
Reviewer 1 Report
Comments and Suggestions for Authors
REVIEW OF THE ARTICLE BY YEONGYUN JUNG ET AL. ENTITLED 'AGING-INDUCED CHANGES IN CUTIBACTERIUM ACNES AND THEIR EFFECTS ON SKIN ELASTICITY AND WRINKLE FORMATION' (microorganisms-3265366)
The authors study age-dependent changes in mechanical skin properties and the skin bacteriome, as well as the relationships between these parameters, with special attention to the common skin bacterium Cutibacterium acnes. They present indices of skin elasticity curves registered by a cutometer — R0, R2, R5, R6, R7 — along with additional physicochemical skin parameters (sebum level, TEWL, relative redness and melanin levels, pH), and wrinkle geometry metrics. Metabolic pathways of skin bacteria and the relative abundances of bacterial species are reconstructed from metagenomic data. Jung et al. speculate on the role of C. acnes in skin ageing. The article falls within the scope of the journal, and the data presented are novel and interesting. Nevertheless, I suggest some revisions to the text.
ABSTRACT
Line 27: It is slightly confusing, as biotin is a vitamin. What distinction are you making between biotin and vitamin metabolism?
INTRODUCTION
Consider enhancing the introduction with recent references (from the last 5 years) regarding the general aspects of the skin microbiome and skin ageing.
Lines 43-44: Notably, the skin also hosts microscopic animals, such as Demodex spp. (10.3389/fimmu.2023.1151527).
Lines 60-61: The sentence is unclear. The skin microbiome refers to all microorganisms present on the skin. They do not accumulate collagen or elastin, which are fibrillar proteins of the skin matrix. Additionally, UV light induces gene expression, oxidative damage, and cancer independently, although correlations may exist between these processes.
Lines 80-86: It is important to mention the adverse effects of C. acnes on the skin, such as acne development, as well as the patterns of bacterial colonisation (10.3390/life14101271). Furthermore, fluorescence-based methods are crucial in studying C. acnes, in addition to 16S rRNA amplicon sequencing.
MATERIALS AND METHODS
Section 2.1: Was the menstrual cycle, a crucial factor for hormonal status, considered in the study?
Lines 122-125: The correct spelling of Köln is Cologne.
Line 150: Did you use the direct or reverse Simpson index?
Line 153: 'Bray-Curtis dissimilarity' should be 'the Bray-Curtis dissimilarity index'. Please add the original reference: 10.2307/1942268.
Line 168: Are you referring to the absolute values of Spearman's correlation coefficient?
RESULTS
Lines 176, 209, 278-279, 324: Species names should be italicised.
Lines 174-186: This part repeats methods. Focus solely on results here.
Table 1: All abbreviations should be explained in the legend (e.g. TEWL, AU).
Table 1: The correct units for TEWL are g/m²/h.
Table 1: R0 is not dimensionless; it has units of mm.
Lines 220-221: The Chao1 index predicts the number of taxa, so it’s more appropriate to present it as a whole number.
Figure 1g, Lines 226-229: Clarify what the Bray-Curtis distances compare (e.g. between which samples).
Why are there 10 predominant species in Figure 1a but 23 species in the LDA analysis (Figure 1e)?
Lines 225, 234, 238, 328: It’s better to refer to the skin bacterial community and bacterial diversity.
Lines 238-241: Repetition of methods.
In the methods, 'r' was used for Spearman's coefficient, whereas 'R' refers to elasticity curve parameters. In subsections 3.3-3.5, 'R' should be changed to 'r' for consistency.
The correlation between sebum levels and C. acnes is vital, as sebum is a major factor influencing microbial diversity and C. acnes abundance (10.3390/life14101271). Discuss whether age is the primary factor shaping bacterial diversity. The correlation between elasticity parameters and sebum levels (10.1111/jocd.12933) should also be discussed.
Subsection 3.3: Why is S. aureus abbreviated while other Staphylococcus spp. are not?
Lines 264-270: This section seems better suited for the discussion.
Lines 282-286: Repetition of methods.
DISCUSSION
Lines 299-300: How do you explain the presence of photosynthesis-related genes?
The correlation between skin mechanical properties, C. acnes, and factors other than age should also be discussed (10.3390/life14101271, 10.1111/jocd.12933). The impact of lifestyle should also be considered.
Line 377: What do you mean? Alpha diversity is not a parameter that can be higher or lower; clarify the intended meaning.
Lines 386-397: Could this be a cross-correlation due to enhanced sebum production having a positive effect on the skin? Sebum often correlates with C. acnes abundance.
Lines 407-410, 419-433: It’s more appropriate to refer to the skin bacterial community and bacterial diversity.
Lines 419-433: One limitation of the study is that eukaryotic members of the skin community (e.g. Malassezia spp.) were not considered (10.1007/s13555-024-01104-4; 10.3390/life14101271).
Line 436: Species names should be italicised.
Author Response
Notes for the paper corrected according to the reviewers
Dear reviewers,
We sincerely thank you for your great efforts in reviewing our paper. According to your and the editor's comments, the paper has been thoroughly corrected. Our detailed point-by-point responses are provided below. Our revisions to the manuscript have been marked using yellow highlights.
Comments for reviewer #1
Line 27: It is slightly confusing, as biotin is a vitamin. What distinction are you making between biotin and vitamin metabolism?
Response: Thank you for your insightful comment. In the KEGG PATHWAY Database, "Biotin metabolism" is categorized under the broader "Metabolism of cofactors and vitamins" pathway, which includes various vitamin metabolism pathways. However, "Biotin metabolism" is presented as a distinct pathway within this category, highlighting its specific role and importance in various biological functions. In our manuscript, we have followed this structure and mentioned biotin metabolism separately to reflect its unique significance, particularly in the context of skin health (lines 26-28).
Consider enhancing the introduction with recent references (from the last 5 years) regarding the general aspects of the skin microbiome and skin ageing.
Response: Thank you for the suggestion. To address the comment, we have incorporated recent studies from the past five years into the introduction. These references provide an updated view on the general aspects of the skin microbiome and its role in skin aging. The newly added literature highlights the increasing microbial diversity observed with aging, as well as shifts in key species like Cutibacterium acnes and Staphylococcus, which can influence skin barrier function and immune response. Additionally, we have included studies that suggest the skin microbiome composition could serve as a biological clock for predicting skin aging, offering new possibilities for targeted anti-aging therapies based on microbial communities (lines 45-49, 60-63).
Lines 43-44: Notably, the skin also hosts microscopic animals, such as Demodex spp. (10.3389/fimmu.2023.1151527).
Response: Thank you for your helpful suggestion. We acknowledge that the skin also hosts microscopic insects, such as Demodex spp., which are important members of the skin ecosystem. We have revised the sentence to include Demodex spp. and the reference provided to reflect a more comprehensive description of the skin's microbial and microscopic community (lines 42-45).
Lines 60-61: The sentence is unclear. The skin microbiome refers to all microorganisms present on the skin. They do not accumulate collagen or elastin, which are fibrillar proteins of the skin matrix. Additionally, UV light induces gene expression, oxidative damage, and cancer independently, although correlations may exist between these processes.
Response: Thank you for your comment. I agree that the original sentence was unclear. The intention was to explain that UV exposure generates reactive oxygen species (ROS) that regulate gene expression related to collagen degradation and elastin accumulation, contributing to photooxidative skin aging and skin cancer. I have revised the sentence to reflect that ROS, generated by UV exposure, plays a role in these processes, rather than implying that the microbiome itself accumulates collagen or elastin (lines 66-68).
Lines 80-86: It is important to mention the adverse effects of C. acnes on the skin, such as acne development, as well as the patterns of bacterial colonisation (10.3390/life14101271). Furthermore, fluorescence-based methods are crucial in studying C. acnes, in addition to 16S rRNA amplicon sequencing.
Response: Thank you for your insightful feedback. I agree that it is important to mention both the positive and adverse effects of C. acnes on the skin. In the revised manuscript, I have included a discussion on how C. acnes, while beneficial to skin health, is also associated with acne development due to its ability to stimulate inflammation and disrupt skin homeostasis. This dual role of C. acnes in skin health and disease is now stated clearly (lines 73-75).
Additionally, I appreciate your suggestion regarding fluorescence-based methods. I have incorporated a mention of these methods in the manuscript, highlighting their importance in studying the spatial distribution and colonization patterns of C. acnes, which cannot be fully captured by 16S rRNA amplicon sequencing alone (lines 82-87).
Section 2.1: Was the menstrual cycle, a crucial factor for hormonal status, considered in the study?
Response: Thank you for your insightful question. We did take the menstrual cycle into account for the female participants in the younger group. Specifically, we ensured that skin sampling and biomechanical measurements were conducted during the follicular phase (10 days after the start of menstruation) to control for hormonal variations. For the older group, all female participants were postmenopausal. Additionally, we measured estradiol levels for all participants to ensure hormonal consistency. Participants with estradiol levels outside the normal reference ranges were excluded. These ranges were as follows: Follicular phase (21–251 pg/mL), Postmenopausal (<10–28 pg/mL), and Male (11–44 pg/mL). We believe this approach allowed us to control for hormonal fluctuations that could potentially impact the study's findings (lines 113-120).
Lines 122-125: The correct spelling of Köln is Cologne.
Response: Thank you for your comment. Courage-Khazaka Electronic GmbH uses the German spelling "Köln" for its location, which is why we initially retained this spelling in the manuscript. However, in accordance with the suggestion, we have revised it to the English version "Cologne" for consistency (lines 135–137).
Line 150: Did you use the direct or reverse Simpson index?
Response: Thank you for your question. We used the direct Simpson index in our analysis. We will clarify this in the revised manuscript to avoid any confusion (line 163).
Line 153: 'Bray-Curtis dissimilarity' should be 'the Bray-Curtis dissimilarity index'. Please add the original reference: 10.2307/1942268.
Response: Thank you for your comment. We have revised the manuscript to change "Bray-Curtis dissimilarity" to "the Bray-Curtis dissimilarity index" as suggested. Additionally, we have included the original reference (10.2307/1942268) in the revised manuscript (lines 170-171).
Line 168: Are you referring to the absolute values of Spearman's correlation coefficient?
Response: Thank you for your question. Yes, we used only the absolute values of Spearman's correlation coefficient in our analysis. In the manuscript, "r > 0.80" refers to the absolute value of the Spearman's correlation coefficient. We have revised the manuscript to clarify this point (line 185).
Lines 176, 209, 278-279, 324: Species names should be italicised.
Response: Thank you for your comment. We have revised the manuscript to ensure that all species names are italicized as required.
Lines 174-186: This part repeats methods. Focus solely on results here.
Response: Thank you for your comment. We agree that the section repeats some of the methods, and we have revised the manuscript to focus solely on the results in this section (lines 191-195).
Table 1: All abbreviations should be explained in the legend (e.g. TEWL, AU).
Response: Thank you for your comment. We have revised the manuscript to ensure that all abbreviations in Table 1, including TEWL and AU, are properly explained in the table legend (Table 1).
Table 1: The correct units for TEWL are g/m²/h.
Response: Thank you for pointing that out. We have revised the manuscript to ensure that the correct units for TEWL are listed as g/m²/h in Table 1.
Table 1: R0 is not dimensionless; it has units of mm.
Response: Thank you for your comment. We have revised the manuscript to correctly indicate that R0 has units of mm in Table 1.
Lines 220-221: The Chao1 index predicts the number of taxa, so it’s more appropriate to present it as a whole number.
Response: Thank you for your comment. We agree that the Chao1 index predicts the number of taxa and should be presented as a whole number. We have revised the manuscript to present the Chao1 values as integers (line 231).
Figure 1g, Lines 226-229: Clarify what the Bray-Curtis distances compare (e.g. between which samples).
Response: Thank you for your comment. In Figure 1g, the Bray-Curtis distances compare the microbial community compositions within the older group and within the younger group, respectively. We have revised the figure legend to clarify this in the manuscript.
Why are there 10 predominant species in Figure 1a but 23 species in the LDA analysis (Figure 1e)?
Response: Thank you for your question. The difference between the number of species in Figure 1a and Figure 1e arises from the distinct nature of the analyses. In Figure 1a, we presented the 10 most predominant species based on their relative abundance across all samples. However, the LDA analysis in Figure 1e identifies 23 significant species based on their differential abundance and ability to distinguish between the two age groups. LDA focuses on species that contribute most to group separation, even if they are not among the most abundant species overall.
Lines 225, 234, 238, 328: It’s better to refer to the skin bacterial community and bacterial diversity.
Response: Thank you for your suggestion. We agree that it would be more appropriate to refer to the terms "skin bacterial community" and "bacterial diversity" in the mentioned sections. We have revised the manuscript accordingly.
Lines 238-241: Repetition of methods.
Response: Thank you for your comment. We acknowledge that the section repeats some of the methods, and we have revised the manuscript to focus solely on the results in this part (lines 248-250).
In the methods, 'r' was used for Spearman's coefficient, whereas 'R' refers to elasticity curve parameters. In subsections 3.3-3.5, 'R' should be changed to 'r' for consistency.
Response: Thank you for pointing this out. We have revised the manuscript to ensure consistency by changing 'R' to 'r' for Spearman's correlation coefficient in subsections 3.3-3.5.
The correlation between sebum levels and C. acnes is vital, as sebum is a major factor influencing microbial diversity and C. acnes abundance (10.3390/life14101271). Discuss whether age is the primary factor shaping bacterial diversity. The correlation between elasticity parameters and sebum levels (10.1111/jocd.12933) should also be discussed.
Response: Thank you for your valuable comment. We agree that there is a significant correlation between C. acnes and sebum levels, as sebum is a critical factor influencing microbial diversity and C. acnes abundance. Additionally, several studies have confirmed that sebum production changes with age. In our results, we observed differences in sebum levels between the older and younger groups, but these differences were not statistically significant. This may be attributed to the fact that we did not account for participants’ skin types during recruitment. We acknowledge that considering skin types in future studies could provide more accurate insights, and we will take this into account in subsequent experiments.
We also analyzed the correlations between sebum levels and C. acnes abundance, as well as the elasticity parameters and bacterial diversity. However, in our data, no statistically significant correlations were found between these variables. Additionally, other factors, such as TEWL, melanin, and erythema were examined for correlations with C. acnes, but none yielded significant results. We have included these non-significant findings in Table S1 as supplementary material.
Subsection 3.3: Why is S. aureus abbreviated while other Staphylococcus spp. are not?
Response: Thank you for your observation. We acknowledge the inconsistency in using the abbreviation for Staphylococcus aureus while other Staphylococcus spp. are not abbreviated. In the revised manuscript, we have used the full abbreviated names for Staphylococcus aureus to maintain consistency throughout the text.
Lines 264-270: This section seems better suited for the discussion.
Response: Thank you for your comment. We agree that this section is better suited for the Discussion section. We have revised the manuscript and moved this part to the Discussion to ensure it aligns with the flow of the results and discussion (lines 408-414).
Lines 282-286: Repetition of methods.
Response: Thank you for your comment. We acknowledge that there was some repetition of the methods in this section. In the revised manuscript, we have removed the redundant explanation of the methods and focus solely on the results in this part.
Lines 299-300: How do you explain the presence of photosynthesis-related genes?
Response: Thank you for your question. The presence of photosynthesis-related genes in the older group could be due to the presence of specific phototrophic microorganisms, such as cyanobacteria. While there was no significant difference, we did observe a higher abundance of cyanobacteria in the older group compared to in the younger group. According to studies such as 10.3389/fmicb.2020.565549, cyanobacteria might influence skin aging. However, since our study focuses more on Cutibacterium acnes rather than cyanobacteria, we did not include this in the manuscript.
The correlation between skin mechanical properties, C. acnes, and factors other than age should also be discussed (10.3390/life14101271, 10.1111/jocd.12933). The impact of lifestyle should also be considered.
Response: Thank you for your insightful comment. We agree that factors other than age, such as lifestyle and sebum production, can influence the correlation between C. acnes and skin mechanical properties. Studies such as 10.3390/life14101271 and 10.1111/jocd.12933 highlight the impact of sebum on microbial diversity and C. acnes abundance, which may in turn affect skin elasticity and wrinkle formation. We have revised the manuscript to include a discussion of these factors and their potential impact (lines 406-413).
Line 377: What do you mean? Alpha diversity is not a parameter that can be higher or lower; clarify the intended meaning.
Response: Thank you for your comment. We agree that "alpha diversity" itself is not a parameter that can be described as higher or lower. We have revised the manuscript to clarify that we observed a greater species richness and diversity in the older group based on alpha diversity indices (lines 379-380).
Lines 386-397: Could this be a cross-correlation due to enhanced sebum production having a positive effect on the skin? Sebum often correlates with C. acnes abundance.
Response: Thank you for your valuable comment. We agree that there is a significant correlation between C. acnes and sebum levels, as sebum is a critical factor influencing microbial diversity and C. acnes abundance. Additionally, several studies have confirmed that sebum production changes with age. In our results, we observed differences in sebum levels between the older and younger groups, but these differences were not statistically significant. This may be attributed to the fact that we did not account for participants’ skin types during recruitment. We acknowledge that considering skin types in future studies could provide more accurate insights, and we will take this into account in subsequent experiments.
We also analyzed the correlations between sebum levels and C. acnes abundance, as well as the elasticity parameters and bacterial diversity. However, in our data, no statistically significant correlations were found between these variables. Additionally, other factors, such as TEWL, melanin, and erythema were examined for correlations with C. acnes, but none yielded significant results. We have included these non-significant findings in Table S1 as supplementary material.
Lines 407-410, 419-433: It’s more appropriate to refer to the skin bacterial community and bacterial diversity.
Response: Thank you for your suggestion. We agree that it is more appropriate to refer to the "skin bacterial community" and "bacterial diversity" instead of "skin microbiome" and "microbial diversity." We have revised the manuscript accordingly to ensure clarity and accuracy in terminology.
Lines 419-433: One limitation of the study is that eukaryotic members of the skin community (e.g. Malassezia spp.) were not considered (10.1007/s13555-024-01104-4; 10.3390/life14101271).
Response: Thank you for your valuable comment. We acknowledge that one limitation of the study is the exclusion of eukaryotic members of the skin community, such as Malassezia spp. While our study focused primarily on the bacterial component of the skin microbiome, we recognize the importance of eukaryotic microorganisms and their potential role in skin biomechanical properties. We have revised the manuscript to include this as a limitation and suggest that future research should consider eukaryotic members, such as Malassezia spp., to provide a more comprehensive understanding of the skin microbial ecosystem (lines 454-459).
Line 436: Species names should be italicised.
Response: Thank you for your comment. We ensure that all species names are italicised in the revised manuscript to follow proper formatting guidelines.

Reviewer 2 Report
Comments and Suggestions for Authors
1. The introduction effectively sets up the context but could benefit from clearly stating the research gap earlier. Consider explicitly highlighting how this study differs from prior studies on the skin microbiome and aging.
2. While the methodology is robust, additional details are needed regarding the participant selection criteria, such as whether other demographic variables (e.g., diet or lifestyle) were considered. This would enhance reproducibility and provide context for the generalizability of the findings.
3. Background descriptions for skin health can be strengthened by citing 10.1016/j.ccr.2023.215426; 10.1021/acsami.1c25014 and what are the advantages of the current work compared to published articles in related fields?
4. The statistical methods used are well-defined, but the rationale for choosing certain tests, such as the Wilcoxon rank-sum test, could be more clearly explained. It would help to justify why specific metrics were used to compare the two age groups.
5. The limitations section acknowledges the sample size and generalizability issues, but it would be valuable to add more discussion on the influence of other potential confounding factors (e.g., diet, cosmetic use, and geographic differences in microbiome composition).
6. The discussion makes good points about the implications of C. acnes in skin health, but the potential mechanisms for why C. acnes might have anti-aging effects could be elaborated more thoroughly. Including more discussion of recent literature on this topic would strengthen the manuscript.
7. The conclusion mentions future research avenues, but it would be more impactful to suggest specific experimental approaches or interventions that could validate the findings, such as microbiome modulation strategies or animal model studies.
Author Response
Notes for the paper corrected according to the reviewers
Dear reviewers,
We sincerely thank you for your great efforts in reviewing our paper. According to your and the editor's comments, the paper has been thoroughly corrected. Our detailed point-by-point responses are provided below. Our revisions to the manuscript have been marked using yellow highlights.
Comments for reviewer #2
1. The introduction effectively sets up the context but could benefit from clearly stating the research gap earlier. Consider explicitly highlighting how this study differs from prior studies on the skin microbiome and aging.
Response: Thank you for your valuable feedback. Based on your suggestion, we have revised the introduction to clearly state the research gap earlier and highlight how our study differs from prior research on skin microbiome and aging. Specifically, we emphasized the limitations of 16S rRNA gene sequencing in previous studies and the lack of objective assessments of skin biomechanical properties. Our study addresses these gaps by incorporating whole-metagenome sequencing for species-level identification and functional profiling, along with objective measurements of skin elasticity and wrinkle formation. We believe these revisions strengthen the introduction by clarifying the study’s unique contributions to the field (lines 92-99).
2. While the methodology is robust, additional details are needed regarding the participant selection criteria, such as whether other demographic variables (e.g., diet or lifestyle) were considered. This would enhance reproducibility and provide context for the generalizability of the findings.
Response: Thank you for your insightful comment regarding participant selection criteria. We acknowledge that demographic variables such as diet or lifestyle were not explicitly controlled in our study. All participants were recruited from the same geographic region (Seoul, South Korea), and the inclusion criteria focused on ensuring that participants were free from chronic skin conditions and had not used antibiotics or steroids in the two weeks prior to the study.
We recognize that factors such as diet, lifestyle, and other demographic variables can significantly impact the skin microbiome. While we did not collect detailed information on these aspects in the current study, we agree that controlling for these variables would enhance the reproducibility and generalizability of the findings. We highlight this as a limitation in the discussion and suggest that future studies should consider controlling for these factors to provide a more comprehensive understanding of how they may influence both the skin microbiome and its correlation with biomechanical properties (lines 454-473) .
3. Background descriptions for skin health can be strengthened by citing 10.1016/j.ccr.2023.215426; 10.1021/acsami.1c25014 and what are the advantages of the current work compared to published articles in related fields?
Response: Thank you for your suggestion. I have reviewed the recommended articles. First, the paper titled "Inorganic–organic hybrid nanomaterials for photothermal antibacterial therapy" explores a novel approach to fighting bacterial infections through photothermal therapy (PTT) using inorganic-organic hybrid nanomaterials. It reviews the design and properties of these hybrid nanomaterials, combining the advantages of inorganic and organic elements to optimize antibacterial effects. The paper discusses current challenges and future directions in developing PTT-based antibacterial treatments using these materials. Second, the paper titled "Dendritic Hydrogels with Robust Inherent Antibacterial Properties for Promoting Bacteria-Infected Wound Healing" introduces a cationic hydrogel (PHCI) with strong inherent antibacterial properties, designed to promote the healing of bacteria-infected wounds. Although both papers offer valuable insights into antibacterial therapies, I couldn't find specific information related to background descriptions for skin health in these studies. If there are particular sections you recommend, I would be happy to reconsider and incorporate them into the manuscript.
4. The statistical methods used are well-defined, but the rationale for choosing certain tests, such as the Wilcoxon rank-sum test, could be more clearly explained. It would help to justify why specific metrics were used to compare the two age groups.
Response: Thank you for your valuable comment. We used the Wilcoxon rank-sum test because it is a non-parametric test that does not assume a normal distribution of the data, making it suitable for comparing the two age groups when the data distributions are unknown or not normally distributed. Since many of the skin biomechanical property measurements did not meet the assumptions of normality, the Wilcoxon rank-sum test was chosen to provide a more robust comparison. We have added this rationale to the manuscript to clarify the justification for using this test (lines 163-166).
5. The limitations section acknowledges the sample size and generalizability issues, but it would be valuable to add more discussion on the influence of other potential confounding factors (e.g., diet, cosmetic use, and geographic differences in microbiome composition).
Response: Thank you for your insightful comment. We agree that the inclusion of a discussion on other potential confounding factors, such as diet, cosmetic use, and geographic differences, would strengthen the limitations section. In response, we have revised the manuscript to address these factors and their possible influence on the study’s findings. Specifically, we have added a discussion on how diet, cosmetic use, and geographic factors can affect the skin microbiome composition and skin biomechanical properties, and emphasized the need for future studies to control for these variables through multivariate analyses. This revision aims to provide a more comprehensive understanding of the limitations of our study and to guide future research (lines 454-473).
6. The discussion makes good points about the implications of C. acnes in skin health, but the potential mechanisms for why C. acnes might have anti-aging effects could be elaborated more thoroughly. Including more discussion of recent literature on this topic would strengthen the manuscript.
Response: Thank you for your thoughtful comment. We agree that the discussion on the potential mechanisms underlying the anti-aging effects of C. acnes could be expanded. To address this, we have revised the manuscript to include a more thorough explanation of the mechanisms through which C. acnes may contribute to skin health and aging. Specifically, we have discussed its role in producing short-chain fatty acids, such as propionic acid, which have been shown to support skin barrier function and exhibit anti-inflammatory properties. Additionally, we have included more recent literature on the immunomodulatory effects of C. acnes and its ability to inhibit oxidative stress, which could further explain its protective role in skin aging. These additions provide a more comprehensive view of C. acnes as a potential anti-aging factor in the skin microbiome.
7. The conclusion mentions future research avenues, but it would be more impactful to suggest specific experimental approaches or interventions that could validate the findings, such as microbiome modulation strategies or animal model studies.
Response: Thank you for your valuable suggestion. We agree that providing more specific experimental approaches would strengthen the conclusion. In response, we have revised the conclusion to suggest potential microbiome modulation strategies, such as probiotic or prebiotic treatments, that could enhance the abundance of beneficial bacteria like C. acnes to maintain skin elasticity and reduce wrinkle formation. Additionally, we propose animal model studies to explore the underlying mechanisms of how microbial communities interact with skin biomechanical properties over time. These experimental approaches would help to validate our findings and further investigate the therapeutic potential of targeting the skin microbiome to improve skin health (lines 469-473).

Round 2
Reviewer 1 Report
Comments and Suggestions for Authors
I have evaluated authors' responses and the revised version of the manuscript. Now it is sufficiently improved. The most of my reccomendations have been taken into account.
PS. In the final version, please, doublecheck the units in Table 1. Should wrinkle depth be in μm?